

# Therapeutic targets and signaling pathways of active components of QiLing decoction against castration-resistant prostate cancer based on network pharmacology

Hongwen Cao[1,*], Dan Wang[1,*], Renjie Gao[1], Chenggong Li[2], Yigeng Feng[1] and Lei Chen[1]

[1] Urology, LONGHUA Hospital Shanghai University of Traditional Chinese Medicine, Shanghai, China
[2] Andrology of Urology, Linshu Hospital of Traditional Chinese Medicine, Linyi, China
[*] These authors contributed equally to this work.

## ABSTRACT

QiLing decoction (QLD) is a traditional Chinese medicine compound. This study aims to explore the therapeutic effect of QLD in castration-resistant prostate cancer (CRPC) and its potential bio-targets. A total of 51 active components and QLD 149 targets were identified using bioinformatics analysis. Additionally, five optimal hub target genes were screened including tumor protein P53 (TP53), interleukin-6 (IL-6), vascular endothelial growth factor-A (VEGF-A), caspase-3 (CASP-3), and estrogen receptor-1 (ESR-1). The interrelated network between active components of QLD and their potential targets was constructed. The molecular function, biological processes, and signaling pathways of QLD-against CRPC were identified. Moreover, QLD was found to efficiently exert a repressive effect on CRPC tumor growth mainly by suppressing the activation of HIF-α/VEGFA and TNF-α/IL6 signaling pathways, and increasing the P53 expression level. These results successfully indicated the potential anti-CRPC mechanism of the active components of QLD.

## INTRODUCTION

Prostate cancer (PCa) is a common malignancy with a poor prognosis and is the second primary cause of malignancy-related mortality worldwide (*Siegel, Miller & Jemal, 2020*). PCa is characteristically resistant to chemotherapy, therefore, androgen deprivation therapy (ADT) has not still shown a full efficacy (*Sulsenti et al., 2021*), and approximately 10–20% of PCa patients may progress to castration-resistant prostate cancer (CRPC) (*Kirby, Hirst & Crawford, 2011*). The prognosis of patients with metastatic CRPC is poor since the androgen receptor (AR) signaling pathways are activated due to drug resistance. The therapeutic effect next generation axis inhibitor AR for the treatment of CRPC is still not satisfactory, despite improvements in this treatment modality (*Qin et al., 2020*).

Corresponding authors
Yigeng Feng, fengyigenglh@sohu.com
Lei Chen, chenlei2114@shutcm.edu.cn

Recently, several studies have reported on the use of chemotherapy for CRPC. For example, the structurally unique androgen-receptor antagonist, darolutamide has been used to delay metastasis and death in CRPC patients (*Fizazi et al., 2019*). Abiraterone acetate has shown promise in the treatment of CRPC (*Attard et al., 2009*), and cabazitaxel has been used as a survival-prolonging treatment (*Pezaro et al., 2014*). Poly ADP ribose polymerase (PARP) inhibition is another widely used drug for CRPC patients. Approximately 25% of metastatic CRPC patients have enriched alterations in DNA damage response genes (*Robinson et al., 2015*). and clinical trials of olaparib and rucaparibs have been efficacious in patients even with deleterious BRCA2 variants (*De Bono et al., 2020*; *Mateo et al., 2020*). Lutetium-177–labeled prostate-specific membrane antigen (PSMA) is a novel agent used to treat CRPC, and may prolong the overall survival of CRPC patients with a lower rate of severe toxicity (*Sadaghiani et al., 2021*). However, the current treatment approaches for CRPC are still limited and there is an urgent need to find pharmacologically bioactive components and targets for CRPC.

QiLing decoction (QLD) is a traditional Chinese medicine compound, consisting of 10 components, including raw astragalus (huangqi), rubescens (donglingcao), codonopsis (dangsheng), turmeric (jianghuang), rehmannia glutinosa (dihuang), shuyangquan, motherwort (yimucao), psoralen (buguzhi), shegan, and licorice (zhigancao). Extensive studies have reported that several components of QLD may be involved in mediating cancer progression. For example, rubescens may inhibit the growth and angiogenesis of breast cancer (*Sartippour et al., 2005*), and astragalus saponins may modulate the progression of colon cancer by regulating glucose-mediated protein expression (*Wang et al., 2014*). Codonopsis lanceolata polysaccharide (CLPS) has been shown to inhibit melanoma metastasis by mediating integrin signaling pathways (*Liu et al., 2017*). These components may also regulate the progression of PCa. For instance, astragalus polysaccharides may inhibit tumorigenesis and lipid metabolism in PCa by mediating the miR-138-5p/SIRT1/SREBP1 pathway (*Guo et al., 2020*); curcumin blocks the progression of PCa by the suppression of the c-Jun N-terminal kinase (JNK) pathway in a epigenetic regulation-dependent manner (*Zhao et al., 2018*); and psoralea corylifolia may promote the apoptosis and autophagy of PC-3 cells in prostate cancer (*Lin et al., 2018*). However, QLD-related biotargets regulating CRPC have not been well studied. Therefore, we aimed to explore the relationships between the active ingredients of QLD and their potential therapeutic targets, and to identify the signaling pathways for CRPC (Fig. 1). In addition, we conducted *in vitro* and *in vivo* experiments to verify the underlying mechanism regulating the therapeutic effects of QLD on CRPC.

## MATERIALS AND METHODS

### Cell culture and reagents

PC3 cells were purchased from the American Type Culture Collection (ATCC; Manassas, VA, USA), and cultured in a Dulbecco's modified Eagle's medium (DMEM) with 10% fetal bovine serum (FBS), 100 units/mL penicillin, and 100 µg/mL streptomycin at 37 °C in a humidified 5% $CO_2$ atmosphere. Formononetin (cat. no. 20100311) was purchased from Jubang Biotechnology Co., Ltd. (Chengdu, China), and Calycosin (20575-57-9)

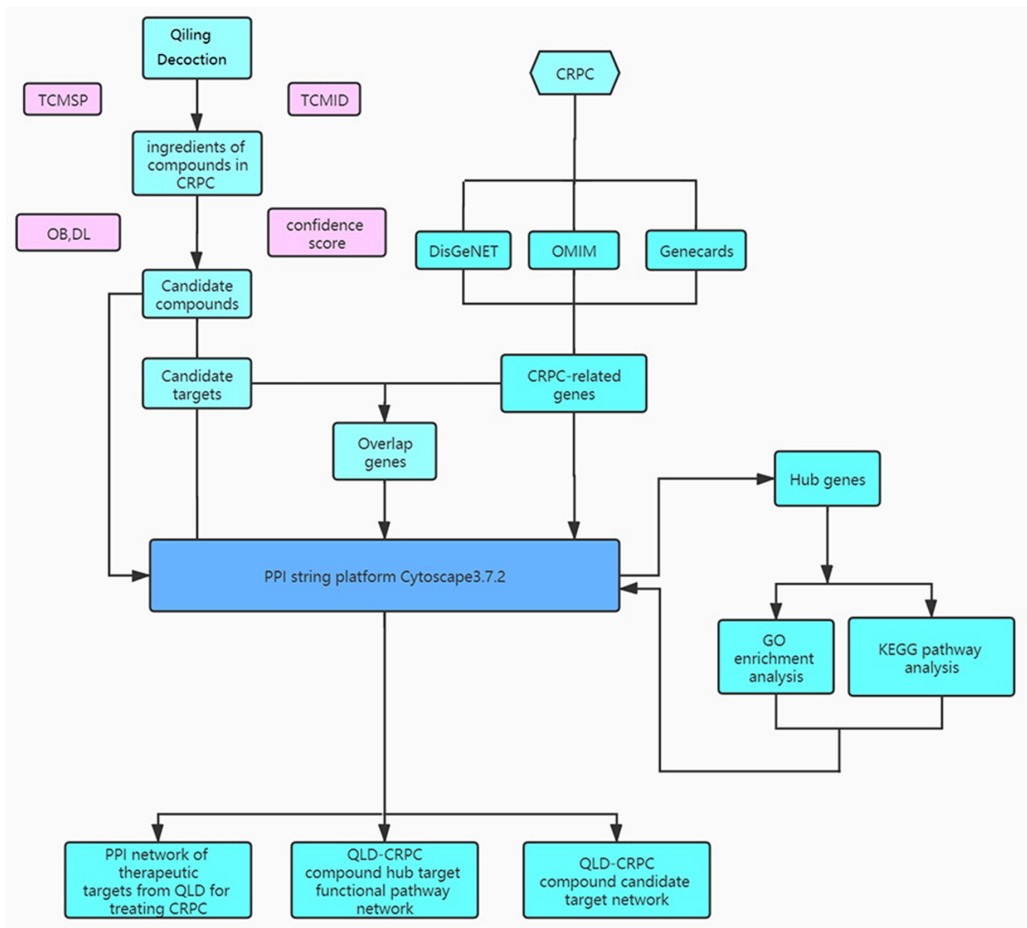

**Figure 1** **The study framework based on network pharmacology.**

was purchased from Sigma-Aldrich (St. Louis, MO, USA), both of which were dissolved with dimethyl sulfoxide (DMSO). The cells in the experimental groups were treated with formononetin (10 μM) or calycosin (20 μM), and the control cells were treated with DMSO.

## Screening of anti-CRPC targets of QLD

The active components of QLD in each herb were identified by the Traditional Chinese Medicine Systems Pharmacology (TCMSP) (http://tcmspw.com/tcmsp.php) and the Traditional Chinese Medicine Integrated Database (TCMID). Oral bioavailability (OB) and drug likeness (DL) were used for the screening of active components, as previously reported. DL ≥ 0.18 and OB ≥ 30% were considered to be the threshold for candidate compounds.

To further identify the target proteins on the active components, we applied the Online Mendelian Inheritance in Man (OMIM; https://omim.org/), DisGeNET (http://www.disgenet.org/), and GeneCards (https://www.genecards.org/) to screen the pathological targets of CRPC. The active components of QLD and potential

pathological targets evaluated were predicted using Venn diagrams. Cytoscape 3.8.2 (https://cytoscape.org/) was used to reveal the active components and potential targets in the form of a network diagram.

## Screening of the most appropriate targets of QLD against CRPC and construction of an interrelated network

The mapped targets of QLD against CRPC were re-analyzed using the STRING database to establish a target-to-target function-related protein network. Cytoscape software (ver. 3.7.1) was used to visualize the results by analyzing the topological parameters (*Su et al., 2014*). The maximum degree-value in the topological data was set as the upper limit and twice the median of degree was taken as the lower limit (*Li et al., 2019*). The top five genes with the highest degree of connectivity were used for screening.

## Gene ontology (GO) and kyoto encyclopedia of genes and genomes (KEGG) pathway enrichment analyses

RStudio 3.6.1 software (http://www.interactivenn.net) was used for the GO and KEGG pathway enrichment analyses to determine the role of target proteins that interact with the active components of QLD in gene function and signaling pathway. The data were acquired using a cut-off value adjusted to $P < 0.05$.

## Establishment of CRPC xenograft models and immunohistochemistry (IHC) *in vivo*

To establish CRPC xenograft models, BALB/c nude mice (six week-old males) were provided by LONGHUA Hospital, which is affiliated with the Shanghai University of Traditional Chinese Medicine (Shanghai, China). The mice were injected with 100 μL phosphate-buffered saline (PBS) containing PC-3 cells into the left axilla of the mice, as described previously (*Fu et al., 2019*). The mice were then randomly divided into four groups ($n = 6$) on day 7 after the injection. The groups included a control group (vehicle solution was administered intragastrically), QLD group (9 g/kg QLD (body weight) was given intragastrically), Doc group (5 mg/kg QLD (body weight) was given intragastrically), and QLD + Doc group (QLD (9 g/kg) and Doc (5 mg/kg) were given intragastrically). The mice with QLD were fed once daily and while the Doc group was fed once a week. The treatment was maintained for 28 days and then all mice were euthanized. The tumor weight and volume were monitored. The tumor volume ($mm^3$) was calculated as follows: tumor volume $= (L \times W^2)/2$ (L: length, W: width). IHC was conducted to detect the expression level of PCNA in cancer cells using PCNA antibody (cat. no. ab29, 1/10000; Abcam, Cambridge, UK), in accordance with previous reports. All applicable international, national, and/or institutional guidelines for the care and use of animals were followed. The protocol was approved by the Ethics Committee of LONGHUA Hospital Shanghai University of Traditional Chinese Medicine.

## Cell counting kit-8 (CCK-8) assay

The PC3 cells were cultured in 96-well plates and when the cells reached a density of $2 \times 10^3$ cells/well, 10% CCK-8 (Sigma-Aldrich) was added into 100 ul fresh mixture, followed by

incubation at 37 °C for 1 h. Finally, the optical density (OD) value was measured at a wavelength of 450 nm using a spectrophotometer.

## Colony formation assay

The PC3 cells treated with the indicated reagents were seeded into 6-well culture plates at a density of $1 \times 10^3$ cells/well. Two weeks later, the colonies were fixed with 4% paraformaldehyde for 15 min at room temperature and visualized by a camera.

## Western blot assay

Total proteins were extracted from cells using RIPA lysis buffer (Beyotime Institute of Biotechnology, Shanghai, China). The proteins were quantified using a BioRad Bradford protein assay kit and electrophoresed into sodium dodecyl sulfate-polyacrylamide gel electrophoresis (SDS-PAGE). Then, the proteins were transferred onto polyvinylidene fluoride (PVDF) membranes, followed by incubation with anti-HIF-$\alpha$ (Abcam, Cambridge, UK; cat. no. ab179483, 1:2000), anti-vascular endothelial growth factor A (VEGFA) (Abcam, Cambridge, UK; cat. no. ab214424, 1:1000), anti-tumor necrosis factor alpha (TNF-$\alpha$) (Abcam, Cambridge, UK; cat. no. ab208156, 1:2000), anti-interleukin 6 (IL6) (Abcam, Cambridge, UK; cat. no. ab233706, 1:2000), anti-P53 (Abcam, Cambridge, UK; cat. no. ab26, 1:2000), anti-cleaved-caspase-3 (cat. no. ab32042, 1:2000), anti-Bax (Abcam, Cambridge, UK; cat. no. ab32503, 1:2000), and anti-glyceraldehyde 3-phosphate dehydrogenase (GAPDH) (Abcam, Cambridge, UK; cat. no. ab8245, 1:1000) at 4 °C overnight, and then washed with Tris Buffered Saline with Tween (TBST) three times. The horseradish peroxidase (HRP)-labeled secondary antibodies (ab205718/ab150117, 1:20000, Abcam) were used for the secondary incubation for 1 h. Finally, imaging was performed by the ChemiScope 6200 Touch Chemiluminescence imaging system (Clinx Science Instruments Co. Ltd., Shanghai, China), and the band intensity was quantified using ImageJ software (National Institutes of Health, MD, USA).

## Statistical analysis

All of the data were analyzed by GraphPad Prism (GraphPad Software Inc., San Diego, CA, USA) and SPSS 22.0 (IBM, Armonk, NY, USA) software, and were presented as mean $\pm$ standard deviation (SD). The $t$-test or one-way analysis of variance (ANOVA) were used to compare differences between groups. $P < 0.05$ was considered to be statistically significant.

# RESULTS

## Components of QLD

QLD consisted of 10 herbs, of which we identified six herbs with 1,157 active components using the TCMSP database. Specifically, there was one component in Baimaoteng, 515 components in Huangqi, 41 components in Jianghuang, 236 components in Shegan, 34 components in Shudihuang, and 330 components in Yimucao (File S1). A total of 51 potential components were screened based on the adopted criteria, including OB $\geq$ 30% and DL $\geq$ 0.18 (Table 1).

**Table 1  A list of the active components in Qiling Decoction.**

| Mol ID | Molecule name | OB (%) | Caco-2 | DL |
|--------|---------------|--------|--------|-----|
| MOL006859 | Volon | 35.42 | −1.12 | 0.63 |
| MOL000211 | Mairin | 55.38 | 0.73 | 0.78 |
| MOL000239 | Jaranol | 50.83 | 0.61 | 0.29 |
| MOL000296 | Hederagenin | 36.91 | 1.32 | 0.75 |
| MOL000033 | (3S,8S,9S,10R,13R,14S,17R)-10,13-dimethyl-17-[(2R,5S)-5-propan-2-yloctan-2-yl]-2,3,4,7,8,9,11,12,14,15,16,17-dodecahydro-1H-cyclopenta[a]phenanthren-3-ol | 36.23 | 1.45 | 0.78 |
| MOL000354 | Isorhamnetin | 49.6 | 0.31 | 0.31 |
| MOL000371 | 3,9-di-O-methylnissolin | 53.74 | 1.18 | 0.48 |
| MOL000374 | 5′-hydroxyiso-muronulatol-2′,5′-di-O-glucoside | 41.72 | −2.47 | 0.69 |
| MOL000378 | 7-O-methylisomucronulatol | 74.69 | 1.08 | 0.3 |
| MOL000379 | 9,10-dimethoxypterocarpan-3-O-$\beta$-D-glucoside | 36.74 | −0.63 | 0.92 |
| MOL000380 | (6aR,11aR)-9,10-dimethoxy-6a,11a-dihydro-6H-benzofurano[3,2-c]chromen-3-ol | 64.26 | 0.93 | 0.42 |
| MOL000387 | Bifendate | 31.1 | 0.15 | 0.67 |
| MOL000392 | Formononetin | 69.67 | 0.78 | 0.21 |
| MOL000398 | Isoflavanone | 109.99 | 0.53 | 0.3 |
| MOL000417 | Calycosin | 47.75 | 0.52 | 0.24 |
| MOL000422 | Kaempferol | 41.88 | 0.26 | 0.24 |
| MOL000433 | FA | 68.96 | −1.5 | 0.71 |
| MOL000438 | (3R)-3-(2-hydroxy-3,4-dimethoxyphenyl)chroman-7-ol | 67.67 | 0.96 | 0.26 |
| MOL000439 | isomucronulatol-7,2′-di-O-glucosiole | 49.28 | −2.22 | 0.62 |
| MOL000442 | 1,7-Dihydroxy-3,9-dimethoxy pterocarpene | 39.05 | 0.89 | 0.48 |
| MOL000098 | Quercetin | 46.43 | 0.05 | 0.28 |
| MOL000449 | Stigmasterol | 43.83 | 1.44 | 0.76 |
| MOL000493 | Campesterol | 37.58 | 1.31 | 0.71 |
| MOL000953 | CLR | 37.87 | 1.43 | 0.68 |
| MOL002322 | Isovitexin | 31.29 | −1.24 | 0.72 |
| MOL001735 | Dinatin | 30.97 | 0.48 | 0.27 |
| MOL000449 | Stigmasterol | 43.83 | 1.44 | 0.76 |
| MOL000351 | Rhamnazin | 47.14 | 0.53 | 0.34 |
| MOL000354 | Isorhamnetin | 49.6 | 0.31 | 0.31 |
| MOL003741 | Anhydrobelachinal | 43.57 | 0.81 | 0.78 |
| MOL003742 | ardisianone A | 44.22 | 0.8 | 0.25 |
| MOL003743 | Belachinal | 31.24 | −0.09 | 0.64 |
| MOL003744 | Belamcandal | 30.07 | 0.05 | 0.67 |
| MOL003753 | Dihydrokaempferide | 50.56 | 0.08 | 0.27 |
| MOL003754 | Epianhydrobelachinal | 43.57 | 0.84 | 0.78 |
| MOL003757 | iristectorene B | 32.56 | 0.36 | 0.42 |
| MOL003758 | Iristectorigenin (9CI) | 71.55 | 0.55 | 0.34 |
| MOL003759 | Iristectorigenin A | 63.36 | 0.54 | 0.34 |
| MOL003769 | Irolone | 46.87 | 0.57 | 0.36 |

| Mol ID | Molecule name | OB (%) | Caco-2 | DL |
|--------|---------------|--------|--------|-----|
| MOL003773 | Mangiferolic acid | 36.16 | 0.66 | 0.84 |
| MOL000006 | Luteolin | 36.16 | 0.19 | 0.25 |
| MOL000359 | Sitosterol | 36.91 | 1.32 | 0.75 |
| MOL000449 | Stigmasterol | 43.83 | 1.44 | 0.76 |
| MOL001420 | ZINC04073977 | 38 | 1.46 | 0.76 |
| MOL001421 | Preleoheterin | 85.97 | 0.46 | 0.33 |
| MOL001422 | iso-preleoheterin | 66.29 | 0.44 | 0.33 |
| MOL000098 | Quercetin | 46.43 | 0.05 | 0.28 |
| MOL001439 | arachidonic acid | 45.57 | 1.2 | 0.2 |
| MOL000354 | Isorhamnetin | 49.6 | 0.31 | 0.31 |
| MOL000422 | Kaempferol | 41.88 | 0.26 | 0.24 |
| MOL001418 | Galeopsin | 61.02 | 0.42 | 0.38 |

## Screening of effective anti-CRPC components of QLD

Determining the effectiveness of QLD in preventing and treating CRPC depends on the synergic interaction between active components and their targets; thus, we evaluated the relationships among the active components and their targets using the compound-target network. The active components of six herbs of QLD and their target proteins related to CRPC were detected using TCMSP and TCMID. The interaction between the active components and target proteins was analyzed using the Cytoscape 3.2.1 software. The results are shown in Figs. 2 and 3.

## Construction of the protein–protein interaction (PPI) network and identification of core target genes

A total of 1,086 pharmacological target genes were identified using the TCMID database based on the 51 primary components found above. The overlapping genes were removed and 149 proteins were identified, which were converted into gene names within the limited species of "Homo sapiens" *via* the UniProtKB. Next, the OMIM, DisGeNET, and GeneCards databases were used to collect the CRPC-related target genes. Venn diagrams were drawn to display the intersected genes of these two target sets, and 66 intersected genes were found. The PPI network was then established to display the intersection of the target genes of QLD against CRPC and CRPC-related biotargets based on the STRING database (Fig. 4). The mapped proteins were imported into Cytoscape in order to filter the core target genes, and five core hub target genes were obtained, including tumor protein P53 (TP53), IL6, VEGFA, caspase-3 (CASP3), and estrogen receptor 1 (ESR1) (Fig. 5).

## Core target genes in GO and KEGG pathway enrichment analyses

The potential pathways of QLD were analyzed to better understand the mechanisms involved in the development of CRPC. GO and KEGG pathway enrichment analyses were conducted using the "clusterprofiler" package in R for the amelioration of CRPC. The GO enrichment and the KEGG pathway analyses were conducted to explore the potential biological processes (BP), cellular components (CC), and molecular functions (MF). A histogram and bubble chart were plotted using the ggplot2 package (version 1.0.2) in the

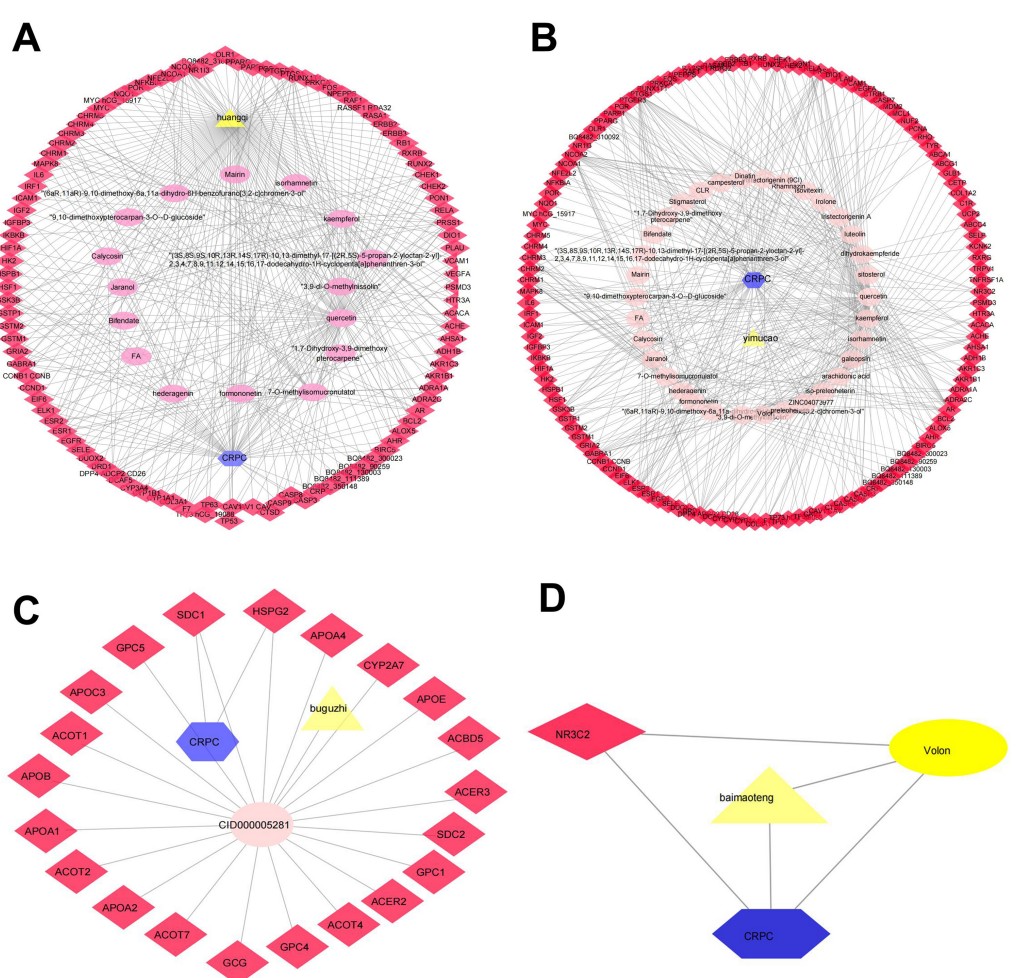

**Figure 2  The QLD-related targets involved in CRPC.** The yellow triangle, red diamond, blue octagon, and the light pink ellipse nodes represent the herbs, targets, disease, and the molecules, respectively. (A) Hunagqi; (B) yimucao; (C) buguzhi; (D) maimaoteng.

R software (Fig. 6). The top 10 terms for GO-BPs were more abundant in the negative mediation of miRNAs production involved in gene silencing by miRNA, gene silencing by miRNA, posttranscriptional gene silencing, gene silencing by RNA, and gene silencing. The regulation of miRNAs production involved in gene silencing was performed by miRNA, and the production of small RNA involved in gene silencing was carried out by dsRNA processing. The remaining BPs are listed in File S2. Additionally, GO-MFs related to the target genes mainly included RNA polymerase II general transcription initiation factor binding, cytokine receptor binding, general transcription initiation factor binding, basal transcription machinery binding, basal RNA polymerase II transcription machinery binding, protease binding, growth factor receptor binding, growth factor activity, cytokine activity, and TFIID-class transcription factor complex binding. The remaining GO-MFs are shown in File S2.

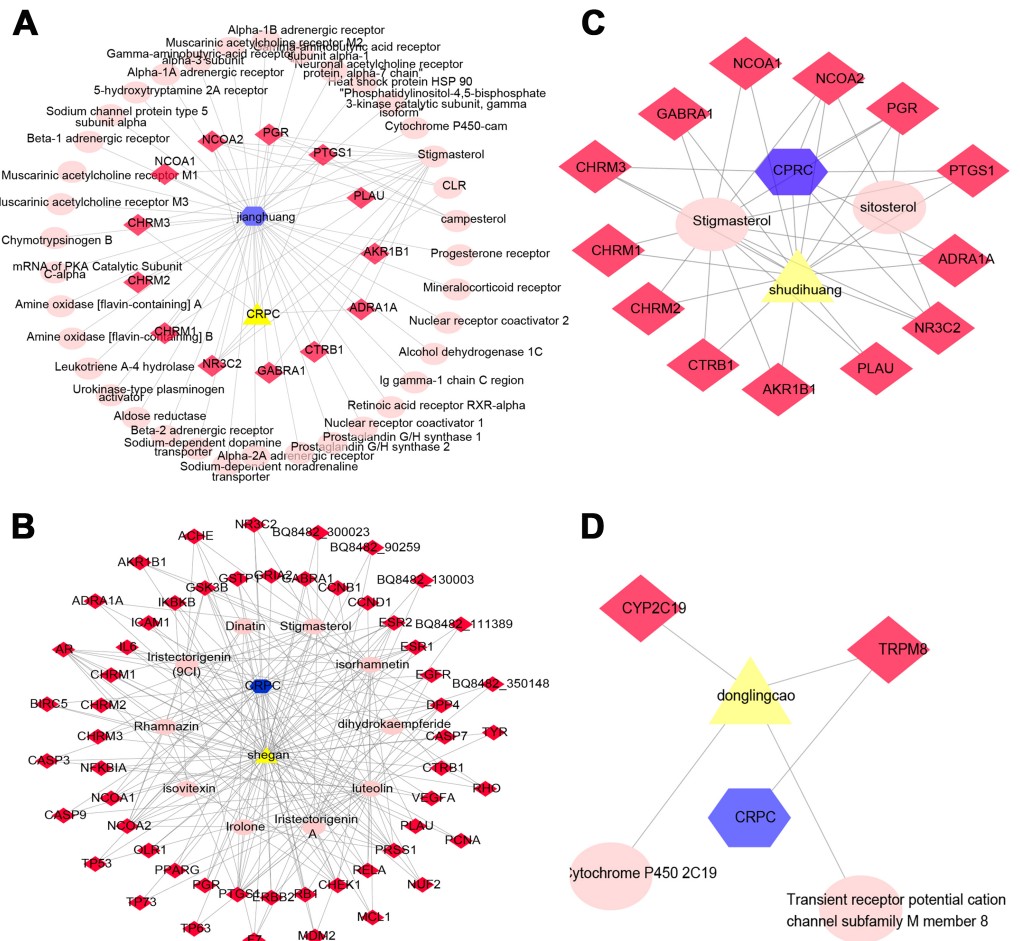

**Figure 3** **The QLD-related targets involved in CRPC. The yellow triangle, red diamond, blue octagon, and the light pink ellipse nodes represent the herbs, targets, disease, and the molecules, respectively.** (A) Janghuang; (B) shegan; (C) shudihuang; (D) dongligncao.

KEGG pathway enrichment analysis demonstrated that the six hub target-related KEGG pathways included the MAPK signaling pathway, resistance to epidermal growth factor receptor-tyrosine kinase inhibitor (EGFR-TKIs), microRNAs in cancer, the PI3K–Akt signaling pathway, AGE–RAGE signaling pathway in diabetic complications, HIF–1 signaling pathway, endometrial cancer, central carbon metabolism in cancer, IL–17 signaling pathway, rheumatoid arthritis, P53 signaling pathway, platinum drug resistance, *etc* (Fig. 7). The remaining pathways are listed in File S3.

### Establishment of network diagram

Cytoscape software was used to determine the targets of QLD against CRPC and the hub target-associated pathways, and a network was constructed. All of the drug protein targets and disease-related proteins were screened. Cytoscape was used to detect the targets of QLD against CRPC, the hub target-associated pathways, and a network visualization of

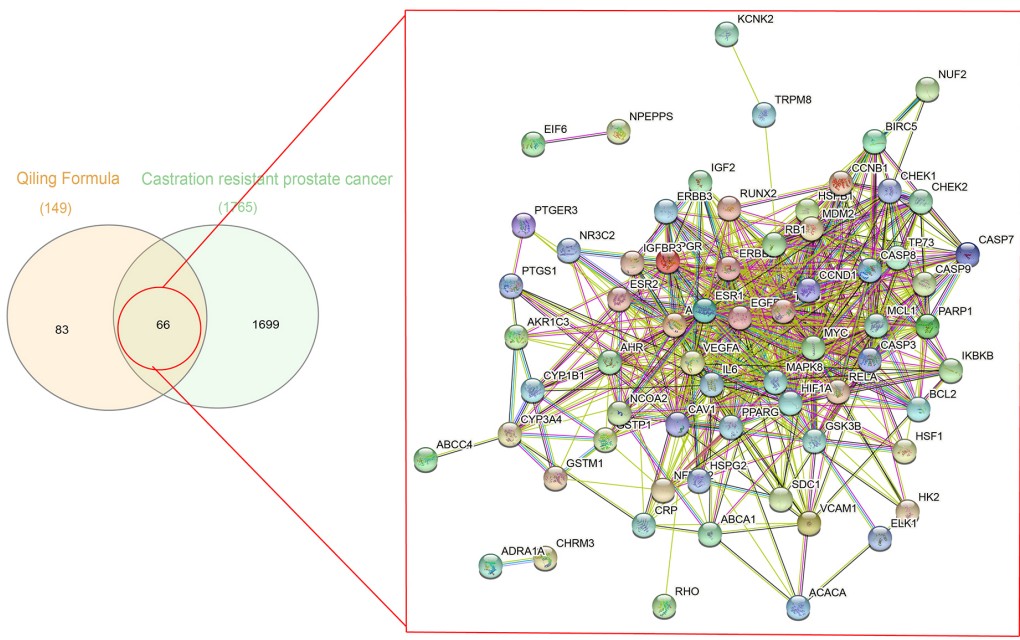

**Figure 4** **A Venn diagram and PPI network were utilized to identify QLD targets involved in CRPC.**
The DisGeNET, GeneCards, and OMIM databases were used based on the inclusion criteria of gene-disease association score > 0.1 and gene score > 1; the high confidence threshold of 0.700 was applied in the STRING database.

QLD against CRPC targets. An interaction diagram for core target-related pathways was plotted (Fig. 8).

## QLD suppressed the growth of CRPC tumor *in vivo* and *in vitro*

We established CRPC xenograft models using PC-3 cells to investigate whether QL has therapeutic effects on CRPC tumor growth *in vivo*. Considering the therapeutic significance of Docetaxel (Doc) on CRPC (*Lowrance et al., 2018*), Doc was chosen as the reference. The xenograft tumor in the QLD group was found to be significantly smaller and lighter than that in the control group. The QLD + Doc group exhibited the lightest and smallest xenograft tumor, with an approximately 76% repression in tumor growth (Figs. 9A–9C). Furthermore, IHC assay demonstrated that QLD efficiently inhibited the expression level of PCNA in the xenograft tumor. The combination of QLD + Doc was found to decrease the level of positive cells *versus* the use of single drugs (Fig. 9D). The *in vivo* assay suggested that QLD could effectively exert repressive effects on CRPC tumor growth. The molecular targets that might be affected by active components of QLD in CRPC were also investigated. The QLD treatment was found to suppress the activity of HIF-a/VEGFA and TNF-a/IL6 signaling pathways and increase the P53 expression level, which is consistent with the results of KEGG pathway enrichment analysis (Fig. 9E). Furthermore, we included two active components, formononetin and calycosin, which have anti-cancer effects on diverse types of cancer (*Yang et al., 2021*; *Yu et al., 2020*). Western blot analysis revealed that formononetin suppressed the growth of CRPC cells by blocking the activity of

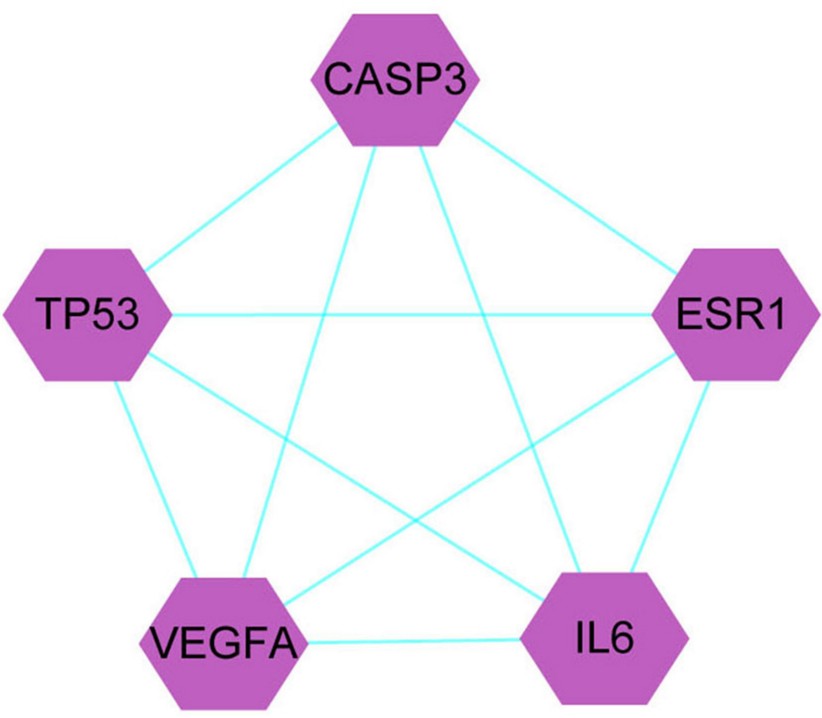

**Figure 5** **The identification of QLD-related hub genes involved in CRPC was performed using Cytoscape software.** The values of 7.055 and 18 were considered to be the median and maximum degrees of freedom, respectively. The top five hub target genes were selected.

HIF-a/VEGFA and TNF-a/IL6 signaling pathways (Figs. 9F–9H). Calycosin also inhibited CPRC cells by inducing apoptosis and activating the P53 pathway (Figs. 9I–9K). QLD was found to efficiently suppress CRPC tumor growth.

## DISCUSSION

CRPC is a life-threatening PCa and studies have reported that CRPC may be ameliorated by the use of medicines, such as niclosamide (*Parikh et al., 2021*) and huaier extract (*Liu et al., 2021*). However, the active components and signaling pathways involved in treatment remain unknown. The single medicinal components in QLD, such as licorice (*Chen et al., 2021*) and rubescens (*Lu et al., 2017*), have been reported to inhibit the progression of PCa by inhibiting the activity of androgen receptors, while the role of pharmacological components of QLD in the treatment of CRPC have not been defined.

We performed a systematic network pharmacology analysis to identify and optimize the screening of biotargets, functional processes, and molecular pathways of QLD against CRPC. Using emerging bioinformatics methods, we obtained biotargets of QLD against CRPC, and the primary core target genes were TP53, IL6, VEGFA, CASP3, and ESR1. The activation of TP53 enhanced the sensitivity of PCa cells against chemotherapeutic drugs and increased the expressions of Raf/MEK/ERK, PI3K/Akt, and DDR1 (*Chappell et al., 2020*). A number of scholars demonstrated that p53-WT suppressed the epithelial–mesenchymal

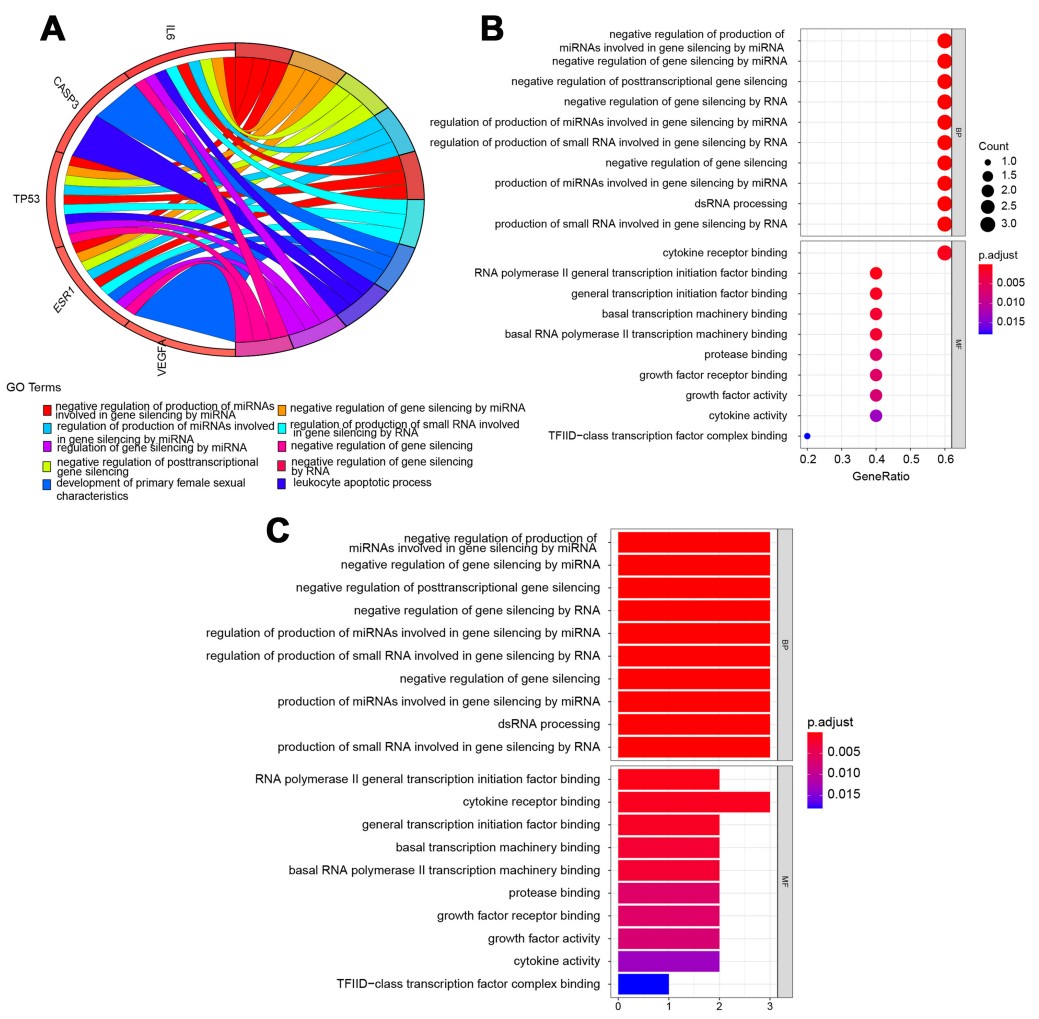

**Figure 6 Biological process (BP) and molecular function (MF) indicated the top 10 BPs and MFs.** The top 10 items were displayed as the (A) blue Circro circles, and 10 BPs and MFs were illustrated as the (B) bubble diagram and (C) histograms.

transition (EMT) process by mediating miR-145 in PCa cells (*Ren et al., 2013*). IL6 was found to affect the proliferation and invasion of PCa cells (*Li et al., 2016*). The VEGF family has been shown to mediate cancer angiogenesis (*Bender & Mac Gabhann, 2015*), and the inhibition of VEGF functionally reduced angiogenesis in PCa cells (*Mu et al., 2020*). However, increased intracellular protein levels of caspase-3 was shown to induce the apoptosis of PCa cells (*Shafiee et al., 2020*). Other research has suggested that ESR1 resulted in the hypermethylation of CpG islands in primary and metastatic human PCa cells (*Wang et al., 2005*; *Yegnasubramanian et al., 2004*). A total of five core target genes were found to mediate CRPC-associated chemotherapeutic sensitivity, EMT, angiogenesis, and cell apoptosis, proliferation, and invasion.

According to the results of the GO annotation analysis, QLD-related pathways against CRPC were mainly enriched in gene silencing by miRNA, posttranscriptional gene silencing,

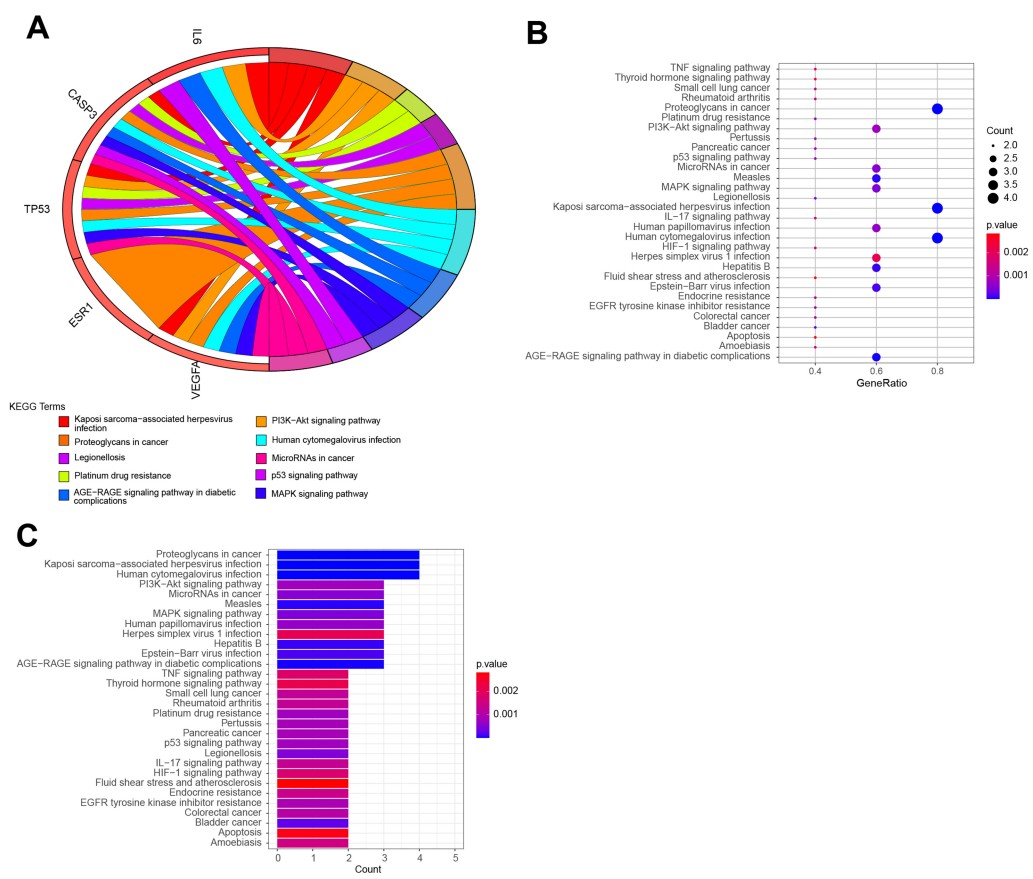

**Figure 7** **The top KEGG pathways of QLD related to CRPC progression.** The top 10 KEGG pathways were exhibited as the (A) blue Circro circles, and (B) bubble diagram and (C) histograms displayed the top 30 KEGG enrichment pathways.

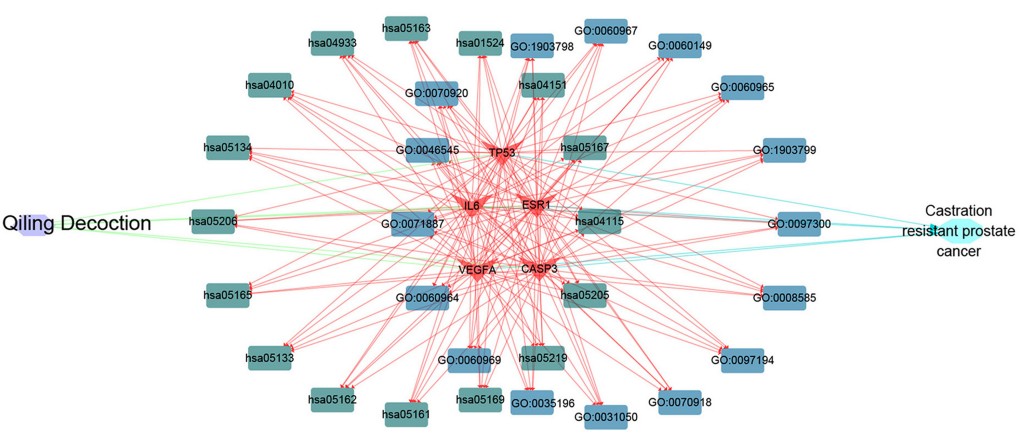

**Figure 8** **The visualization of QLD-target-GO-KEGG-CRPC based on bioinformatics analysis.**

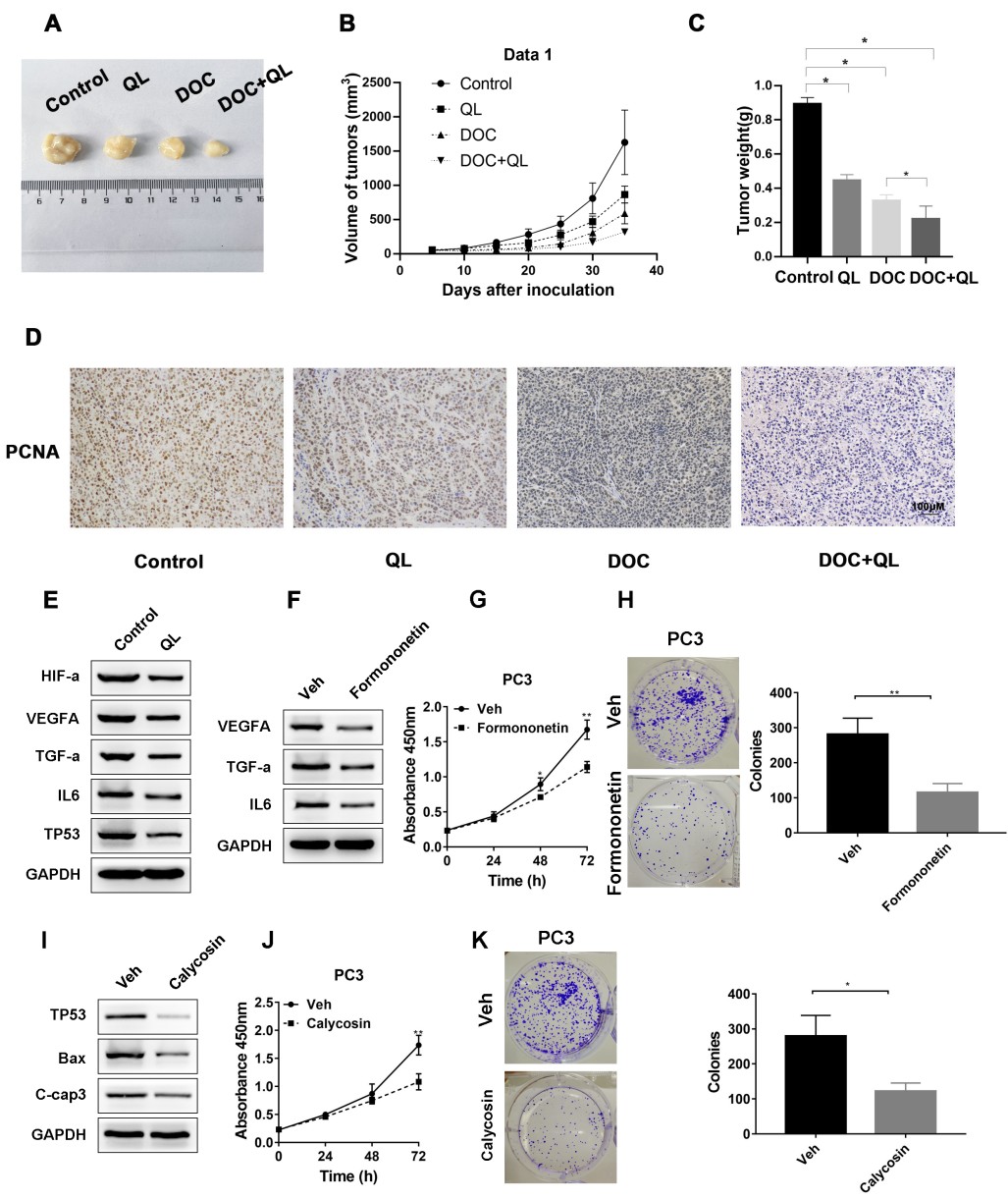

**Figure 9 QLD repressed the growth of CRPC *in vivo* and *in vitro*.** In the *in vivo* experiment, (A) tumor growth was observed, and the (B) volume and (C) weight of tumors were measured; (D) IHC assay was employed to detect the PCNA expression in xenograft tumor. (E) Western blotting was applied to detect the expression levels of HIF-α, VEGFA, TNF-α, IL-6, and P53 in tumor cells treated with QLD and control. (F) PC3 cells were serum-starved, and treated with formononetin for 24 h. Western blotting was employed to detect the expression levels of VEGFA, TNF-α, and IL-6. (G and H) PC3 cells were serum-starved, and treated with formononetin for the indicated time points. (G) CCK-8 and (H) colony formation assays were applied to assess the proliferation of PC3 cells. (I) PC3 cells were serum-starved, and treated with calycosin for 24 h. Western blot and colony formation assays were employed to determine the expression levels of p53, Bax, and C-caspase-3. (J and K) PC3 cells were serum-starved, and treated with calycosin for the indicated time points. (J) CCK-8 and (K) colony formation assays were applied to detect the proliferation of PC3 cells.

cytokine receptor binding, and RNA polymerase II general transcription initiation factor binding. KEGG pathway enrichment analysis revealed that MAPK, EGFR-TKIs, miRNAs, PI3K–Akt, and AGE–RAGE and HIF–1 signaling pathways were enriched. It has been reported that miRNAs are involved in CRPC progression by the suppression of downstream proteins. For example, miR-655 may suppress the proliferation of PCa cells and tumor metastasis by inhibiting TRIM24 (*Bai et al., 2021*) and miR-30a may inhibit the androgen-independent growth of PCa cells by targeting the expression levels of MYBL2, FOXD1, and SOX4 (*Li et al., 2020*). VEGF commonly responds to changes in micro-environment signals such as HIF-1 and the targeting of VEGF pathways in PC is currently being studied (*McKay et al., 2016*). Formononetin, a component of QLD, has been shown to suppress the growth of CRPC cells by blocking the HIF-a/VEGFA signaling pathways, which is consistent with the results from KEGG pathway enrichment analysis. Calycosin was also reported to induce the apoptosis of osteosarcoma mainly through the up-regulation of TP53 and CASP3 expression (*Tan et al., 2020*). The *in vivo* and *in vitro* assays confirmed that QLD may upregulate the expression of p53.

## CONCLUSIONS

In summary, the bioinformatics findings indicated that gene silencing by miRNA and cytokine receptor binding, as well as the targeting of TP53, IL6, VEGFA, CASP3, and ESR1 may be potential pathways of QLD against CRPC. We were also able to identify the tumor inhibitory effects of QLD on CRPC.

### Funding

This work was funded by the General Program of the National Natural Science Foundation of China: Study on the mechanism of Qi Ling Decoction's regulation of JNK/Bcl-2-Beclin1 signal axis via miRNA-143 to inhibit autophagy against Abiraterone resistance in prostate cancer, Subject number: 82174199: Purchase of experimental materials; Shanghai University of Traditional Chinese Medicine Industry Development Center Medical and Nursing Integrated Science and Technology Innovation Project: Study on the mechanism of "Qi Ling Decoction" delaying castration resistance in prostate cancer based on IL6/STAT3-mediated immune pathway in tumor microenvironment, Subject number: 2069: Equipment leasing; Shanghai Municipal Health Commission special subject of Chinese traditional medicine research: Study on the mechanism of the prescriptions of "Qi Ling" regulating tumor microenvironment and inhibiting CRPC cell proliferation and invasion through IL6/STAT3, Subject number: 2020JQ002 and Shanghai Science and Technology Commission Shanghai Natural Science Foundation Project: Study on mechanism about prescriptions of "Qi Ling" inhibiting androgen-independent transformation of prostate cancer cells by AR signaling pathway based on TRIM66 / HP1 gamma complex, Subject number: 19ZR1458200: Testing and processing. The funders had

no role in study design, data collection and analysis, decision to publish, or preparation of the manuscript.

## Grant Disclosures

The following grant information was disclosed by the authors:

General Program of National Natural Science Foundation of China.

Shanghai University of Traditional Chinese Medicine Industry Development Center Medical and Nursing Integrated Science and Technology Innovation Project.

Shanghai Municipal Health Commission special subject of Chinese traditional medicine research.

Shanghai Science and Technology Commission Shanghai Natural Science Foundation Project.

## Competing Interests

The authors declare there are no competing interests.

## Author Contributions

- Hongwen Cao conceived and designed the experiments, prepared figures and/or tables, authored or reviewed drafts of the article, and approved the final draft.
- Dan Wang conceived and designed the experiments, performed the experiments, prepared figures and/or tables, and approved the final draft.
- Renjie Gao analyzed the data, prepared figures and/or tables, and approved the final draft.
- Chenggong Li conceived and designed the experiments, performed the experiments, prepared figures and/or tables, and approved the final draft.
- Yigeng Feng analyzed the data, authored or reviewed drafts of the article, and approved the final draft.
- Lei Chen performed the experiments, authored or reviewed drafts of the article, and approved the final draft.

## Animal Ethics

The following information was supplied relating to ethical approvals (i.e., approving body and any reference numbers):

All applicable international, national, and/or institutional guidelines for the care and use of animals were followed. The protocol was approved by the Ethics Committee of LONGHUA Hospital Shanghai University of Traditional Chinese Medicine.

## Data Availability

The raw data are available in the Supplemental File.

## Supplemental Information

Supplemental information for this article can be found online at http://dx.doi.org/10.7717/peerj.13481#supplemental-information.

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
