# Peer review of "Therapeutic targets and signaling pathways of active components of QiLing decoction against castration-resistant prostate cancer based on network pharmacology"

_PeerJ, doi:10.7717/peerj.13481_

## Round 0.1 · original submission · Major Revisions

Dear Authors,

Please change the manuscript according to the reviewers' comments.

Thanks

best regards

ferdinand frauscher

Reviewer 1 ·

Basic reporting

The author Cao, et al. is using the systematic network pharmacology through various datasets to identify the active ingredients of QiLing Decoction (QL) and the potential molecular targets of QL for castration-resistant prostate cancer (CRPC). Several interesting targets, such as TP53, VEGFA, IL-6, ESR1, and CASPASE 3, have been identified and might be involved in QL targeting CRPC. In addition, biological experiments showed that QL inhibits human prostate cancer cell PC3-derived tumor growth in vivo.

The bioinformatic studies provided some novelty regarding the QL-active components inhibit CRPC through potential targets/signaling pathways. However, there is a lack of biological experiments to validate the function of identified molecular targets that might be affected by the active ingredients of QL in CRPC.

Note: there are many mistakes of grammars, writings/spellings, and repeats (e.g., line 177-178) in the whole text. I suggest you have a colleague who is proficient in English and familiar with the subject matter review your manuscript or contact a professional editing service.

Experimental design

1. Although core targets like TP53, IL-6, VEGFA, Casp3 and ESR1 have been identified, but whether QL active components inhibit tumor growth through targeting one or more of the five top candidates have not been addressed; Similarly, the identified pathway from datasets using computational bioinformatics, but whether and how one of these pathways contribute to QL-mediated CRPC tumor growth are not addressed at all.

The authors showed that in Fig.8. that QL can inhibit PC3-derived CRPC tumor growth in xenograft models. However, what is the core active component from QL? What is the core target for the active ingredients in CRPC? Some targets-based Western and/or RT-PCR could be done to give some answers.

2. Composite ingredients of QI (starting from line 145-148)
The ingredients of the six herb medicines identified from TCMSP database are not from the lists in the 10 individuals of QL (line 77-79). Please explain how these ingredients can be used for QL in your studies.

Validity of the findings

To increase the biological significance of the studies and validate the targets identified from system bioinformatics, one active ingredient from QL and one molecular target in CRPC at least should be verified through experimental studies.

Additional comments

Minor concerns:
1. From line 63 to 76 (Introduction), the authors briefly described how CRPC progress/development and current treatment strategies for CRPC. However, they should give updated and precise background introduction. For example, how androgen receptor (AR) signaling plays an important role in CRPC even in the castration status, FDA approved treatment like PARP inhibitors for CRPC carrying deficiency of for DNA repair, radioligand-based Lu177-PSMA treatment, anti-AR signaling inhibitor Enzalutamide for CRPC, and Taxens-based chemotherapy and more.

2. Please present detailed descriptions in Materials and Methods.

3. In Results section
Figure 2A-D. Labelling words are too small to be recognized, same as in Figure 3 and Figure 5A;

Reviewer 2 ·

Basic reporting

Language editing is highly recommended, including grammar and sometimes to ensure clarity. I have listed some obvious examples here, but the rest of the text needs a touch by a native speaker too.
Line 70: “there are several reports have been reported as for the chemotherapy for CRPC.” The sentence seemed to have redundance description. I suggested to write as “Several studies have reported chemotherapy for CRPC”.
Line 73: “Abiraterone Acetate has been prove to played … …”, which mixed the past tense and present tense.
Line 77: “....consisting of10 single....”. There is one space missing between “of10”.
Line 82: “....by regulatingglucose-regulated....” also has a space missing.
Line 83: there is an additional “c” in the sentence “....c and codonopsis....”.
Line 90: “psoralea corylifolia could caused the apoptosis”, erroneously used the past tense.
Line 95: it seemed that an additional “t” was inserted.

Experimental design

In general, the research question and the methods are reasonable but more details of the methods needed to be added. The “PCOS-Related Target Network” part, for example, the authors only listed three database and criteria for one database. This is too simple. I suggested to add a brief description to state how to establish the network or cite related reference.

More comments are listed below:
Line 102-103: the authors have used three criteria (OB, DL, and Caco-12 in the manuscript. It is puzzling that you listed “bbb” here but not used it.
Line 107: What is TCMID database? Please give the full name and state the reason that you choose this database.

Validity of the findings

No comment.

Additional comments

Line 150: “51 potential ingredients were screened and were displayed in Table 1”. Based on the three criteria, it is better to give a short description of these 51 ingredients, like active ingredients, potential anti-CRPC effect, etc. The word “screened” was also not properly used here. It looks like you only screened 51 compounds. You screened 1,157 compounds and found 51 potential hits.
Line 147: The term “Suangqi” was not consistent with the legend of figure 2 “hunagqi”.
Line 155-156: the authors mentioned that they established a network of 51 ingredients with their targets. That is too ambiguous. You need to give a description of the figures. At least, you may descript one of them as a sample.
Line 164: “PPI network was established to displayed……”, erroneously used the past tense.
Line 171-172: “Next, depending on……involving ameliorating CRPC”. The sentences need to be refined.
Line 171 and 187: here the authors mentioned 6 core/hub targets were tested. But the PPI network identified 5 hub targets. Is there someone missing?
Line 197: please give a detail description of the figure 7. You do not want the readers to figure out the meaning by themselves.
Line 200: “CRPCxenograft” has a space missing.
Line 226: “……p53-WT could resulted in inhibitory affect……”, erroneously used the past tense.

---

## Round 0.2 · Minor Revisions

Dear Authors,

Congratulations on the excellent revised manuscript.

I m very happy to inform you that the manuscript is almost ready to be accepted for publication in PeerJ.

Before that can happen, have the manuscript edited for typos and English language.

Best regards

Ferdinand Frauscher

---

## Round 0.3 · accepted · Accept

I am very happy to inform you that your manuscript is accepted for publication.

best regards

Ferdinand